# AugUndo: Scaling Up Augmentations for Unsupervised Depth Completion

## Abstract

Unsupervised depth completion methods are trained predominantly using structure-from-motion. The training objective involves minimizing photometric reconstruction error between temporally (from video) or spatially (from stereo) adjacent images, which assumes photometric consistency in co-visible regions across frames. Block artifacts from geometric transformations, intensity saturation, and occlusions are amongst the many undesirable by-products of common data augmentation schemes that affect reconstruction quality, and thus the resulting model performance. Hence, typical data augmentations on the image that are viewed as essential to training pipelines in other vision tasks have seen limited use beyond small image intensity changes and flipping. In fact, the sparse depth modality have seen even less variety as intensity transformations alter the scale of the measured 3D scene, and geometric transformations may decimate the sparse points during resampling. We propose a method that unlocks a wide range of previously-infeasible geometric augmentations for unsupervised depth completion. This is achieved by reversing,or "undo"-ing, geometric transformations to the coordinates of the output depth, warping the depth map back to the original reference frame. This enables computing the photometric reprojection loss via the original images and sparse depth maps, eliminating the pitfalls resulting from naive loss computation on the augmented inputs. This simple yet effective strategy allows us to scale up augmentations to boost performance. We demonstrate our method on indoor (VOID) and outdoor (KITTI) datasets where we improve upon three existing methods by an average of 10.4% overall across both datasets.

## 1 Introduction

Data augmentation plays a crucial role in easing the demand for training data as it enriches the amount of data by orders of magnitude, boosting performance on various deep learning tasks (Perez & Wang, 2017; Shorten & Khoshgoftaar, 2019; Taylor & Nitschke, 2018), particularly by preventing overfitting to the training data. Despite its extensive application to computer vision, there is no principal guideline for selecting effective augmentations. The choice of augmentations is often *ad hoc*, and tailored to the specific vision task case-by-case. One common axiom of choosing augmentation is that the task output should remain invariant to the augmentation. For example, image flips are viable augmentations for classification tasks, since it does not alter the resulting object label. Conversely, flipping road signs can alter their meanings, hence such augmentations can be detrimental to tasks involving road sign recognition. For vision tasks aiming to recover geometric information of the underlying 3D scene, consequently, the range of viable augmentations is more restricted. A naive implementation of common image augmentations, such as resizing, flipping, and rotating, alters the scale and directional properties of the image, compromising the assumption that model output should remain invariant. Hence, it is unsurprising that recent research on geometric tasks (Godard et al., 2019; Lyu et al., 2021; Zhang et al., 2023a; Wong et al., 2020; 2021a; Wong & Soatto, 2021) primarily relies on simple photometric augmentations.

Nevertheless, geometric tasks inherently contain physical world priors that may be effectively employed for augmentation purposes. A simple example is random resizing. Zooming an object in the image domain implies a closer distance between the object and the camera, thus naive resizing is not a viable augmentation for depth estimation. However, considering the standard pinhole camera model, the object's projected size in the image is directly proportional to its depth. The ratio between

them is determined by the object size in the physical world and the focal length of the camera, both of which are fixed by their nature. By establishing this relationship, it becomes possible to *un-do* the augmentation in the output space by aligning the training target with the augmented input, Similarly, such alignment in the output space is also feasible when applying other geometric transformations, such as rotation, translation, flipping, etc.

By incorporating the aforementioned output space alignment, we conduct a comprehensive study to explore the best combinations of augmentations. Given the impracticality of exhaustively examining all geometric tasks, we focus on *unsupervised depth completion*, the task of inferring a dense depth map from a single image and its associated sparse depth map. In this task, the set of augmentations have traditionally been restricted to a limited range of photometric transformations and flipping – due to the need to preserve photometric consistency across a sequence of video frames used during training, and the sparse set of 3D points projected onto the image frame as a 2.5D range map. Block artifacts, loss during resampling, intensity saturation are just some of the many undesirable side-effects of traditional augmentations to the image and sparse depth map. Moreover, as unsupervised depth completion (as well as other geometric tasks) rely on reconstruction of observations to compute loss, it is critical that the output and the observation align spatially.

We introduce AugUndo, an augmentation framework that allows one to apply a wide range of photometric and geometric transformations on the inputs, and to "undo" them during the loss computation. This allows one to compute an unsupervised loss on the original images and sparse depth maps, free of artifacts, through a warping of the output depth map – obtained from augmented input – onto the input frame of reference based on the inverse geometric transformation. In addition to group transformations that allow for output alignment, we combine them with commonly employed photometric augmentations. Lastly, we study whether non-group transformations, such as cropping and occlusion, can further improve model performance. We demonstrate AugUndo on three recent unsupervised depth completion methods and evaluate them on indoor and outdoor settings, where we improve by an average of 10.4% across all methods and datasets.

Our contributions are as follows: (i) We propose AugUndo to incorporate a wide range of photometric and geometric augmentations into unsupervised depth completion; AugUndo can be applied in a plug-and-play manner to existing methods with negligible increase in computational costs during training. (ii) While the inverse of group transformations are well-defined, we further define a notion of an "approximate inverse" for each non-group transformation that we apply. By simply applying the (pseudo-)inverse transformation on the output depth, we can align it with the original input reference frame for loss computation, enabling us to scale up augmentations to include a range of geometric, group and non-group, transformations. (iii) Our approach is not limited to depth completion methods, but is also applicable to unsupervised monocular depth estimation. We show that both unsupervised depth completion and estimation methods can be consistently improved with better generalization.

## 2 RELATED WORK

**Data Augmentation for Monocular Depth Completion and Estimation.**    The literature on monocular depth completion and estimation is extensive and we highlight some of the representative work. While it is natural to apply photometric transformations such as intensity and contrast in depth completion (Ma et al., 2019; Wong et al., 2021a;b; 2020) and depth estimation (Guizilini et al., 2020; Poggi et al., 2020; Ranftl et al., 2020; Zhan et al., 2018), geometric data augmentation is less adopted and mostly applied to supervised training. For example, Cheng et al. (2018; 2020); Park et al. (2020); Van Gansbeke et al. (2019) used random scaling, in-plane rotation for supervised depth completion, while Eigen et al. (2014); Eigen & Fergus (2015) additionally used translation for depth estimation. Other works in depth estimation (Eigen et al., 2014; Eigen & Fergus, 2015; Laina et al., 2016; Li et al., 2015; Liu et al., 2015; Ranftl et al., 2020; 2021; Yin et al., 2019; Xu et al., 2017) also adopted similar augmentations; however, for depth completion Hu et al. (2021); Lin et al. (2022); Kam et al. (2022); Yang et al. (2019); Zhang et al. (2023b); Xu et al. (2019) limited their augmentations to color jitter and horizontal flips – this is largely because rotation and scaling would decimate the sparse depth maps causing points to be interpolated away. Nonetheless, for supervised methods, it is straightforward to directly apply the same transformation to the ground truth annotations; we posit that artifacts caused from transformation of a piece-wise smooth depth map are less severe than those of an RGB image and its intensities. These artifacts would in turn affect the training signal, which

relies on photometric correspondences, for unsupervised methods. In contrast, our approach enables diverse geometric augmentations to be applied in a plug-and-play fashion.

**Unsupervised Monocular Depth Completion** assumes stereo images or monocular videos to be available during training. Both stereo (Shivakumar et al., 2019; Yang et al., 2019) and monocular (Ma et al., 2019; Wong et al., 2021a;b; 2020) training paradigms leverage photometric reprojection error as a training signal by minimizing photometric discrepancies between the input image and its reconstruction from other views. In addition to the photometric reconstruction term, depth completion methods also minimize the difference between the input sparse depth map and the predicted depth (where sparse depth points are available) and a local smoothness regularizer. Ma et al. (2019) used Perspective-n-Point (Lepetit et al., 2009) and RANSAC (Fischler & Bolles, 1981) to align consecutive video frames. Yang et al. (2019) learned a depth prior conditioned on the image by pretraining a separate network on ground truth from an additional dataset. Lopez-Rodriguez et al. (2020) also used synthetic data, but applied image translation to obtain ground truth in the real domain. Wong et al. (2020) used synthetic data to learn a prior on the shapes populating a scene. Wong et al. (2021b) proposed an adaptive optimization scheme to reduce penalties incurred on occluded regions. Wong & Soatto (2021) proposed a calibrated back-projection network that introduces an architectural inductive bias by mapping the image onto the 3D scene. Liu et al. (2022) introduced monitored distillation for depth completion. Amongst all of the methods discussed, augmentations are limited to a small range of photometric perturbations and image flipping because operations such as cropping reduces co-visible regions while others that require interpolation (e.g. rotation, resizing) create artifacts, which affects the reconstruction quality and cause performance degradations. Sparse depth maps are further exacerbated as resampling and interpolation may cause loss of sparse points. Contrary to their augmentation schemes, our work allows for a large range of photometric and geometric augmentations to be introduced during training.

**Unsupervised Monocular Depth Estimation.** The common unsupervised/self-supervisory signal comes from spatial or temporal relationships between input images. Garg et al. (2016) proposed to frame depth estimation as a novel view synthesis problem and minimized a photometric loss between the input image and its reconstructions. Godard et al. (2017) improves Garg et al. (2016) by imposing a consistency loss on the depth predicted from left and right images. Zhou et al. (2017) then proposed a pose network to enable self-supervised training on monocular image sequences. Godard et al. (2019) addressed the problem of occlusion and moving objects using auto-masking and min reprojection loss, establishing a strong baseline. To improve the supervision signal, Tosi et al. (2019); Watson et al. (2019); Choi et al. (2021) leverage noisy proxy labels and trinocular assumption (Poggi et al., 2018). Additional loss terms based on visual odometry (Wang et al., 2018), iterative closest point (Mahjourian et al., 2018), surface normals (Yang et al., 2018b), and semantic segmentation (Guizilini et al., 2019; Kumar et al., 2021) were also introduced; Poggi et al. (2020); Yang et al. (2020) further included predictive uncertainty. To handle rigid and non-rigid motions in the scene, previous works explored multi-task learning to include optical flow and moving object estimation (Chen et al., 2019; Ranjan et al., 2019; Yang et al., 2018a; Yin & Shi, 2018; Zou et al., 2018), and used semantics to filter out outlier regions (Klingner et al., 2020). Following this line of works, different network architectures (Lyu et al., 2021; Zhang et al., 2023a; Zhao et al., 2022) were proposed to increase accuracy: Lyu et al. (2021) redesigned the skip connection and decoders to extract high-resolution features, while Zhao et al. (2022) combined global and local representations and Zhang et al. (2023a) introduced a lightweight architecture with dilated convolution and attention. In addition to depth completion, our method also shows consistent improvements on depth estimation for three representative architectures (Godard et al., 2019; Lyu et al., 2021; Zhang et al., 2023a).

## 3    METHOD FORMULATION

Let $I : \Omega \subset \mathbb{R}^2 \to \mathbb{R}^3_+$ be an RGB image that is taken by a calibrated camera. Additionally, paired with each image is a sparse point cloud projected onto the image plane (depth map) $z : \Omega_z \subset \Omega \to \mathbb{R}_+$. Let $K \in \mathbb{R}^{3 \times 3}$ be the intrinsic calibration matrix , either given or estimated. Given a monocular image and its associated sparse depth map, depth completion aims to learn a mapping $\Omega \times \Omega_z \to \mathbb{R}_+$ that recovers the distance between the camera to points in the 3D scene which has been projected to the image domain $\Omega$. In another mode, if sparse depth maps are not given, then the problem reduces to monocular depth estimation which learns a map from a single image to a depth map $\Omega \times \Omega_z \to \mathbb{R}_+$.

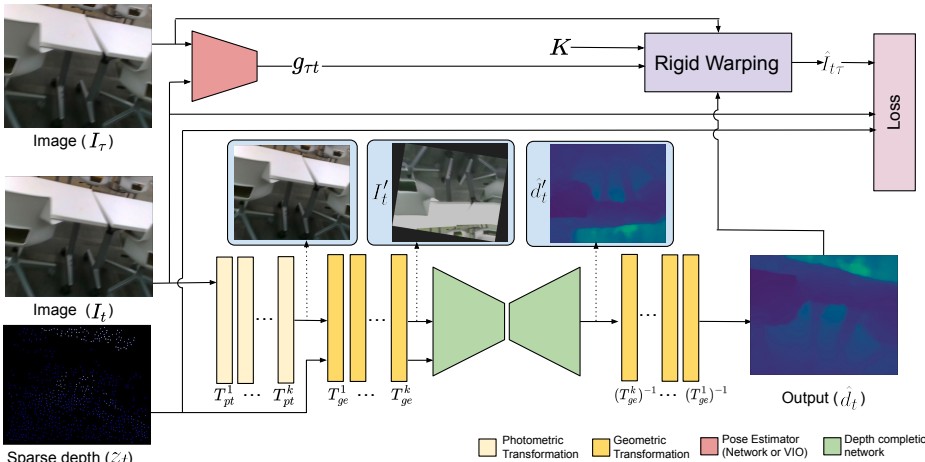

Figure 1: *Overview of the AugUndo.* We augment the input images with photometric transformations and then geometric transformations and augment the input sparse depth with the same set of geometric transformations. Reverse geometric transformations are applied to the output depth to return it to the original reference frame by inverse warping. This enables image and sparse depth reconstruction loss functions to be computed using the original image and sparse depth maps, and minimizes artifacts that have limited the use of extensive augmentations in existing works.

For the ease of notation, we will refer to the output depth map as $d \in \mathbb{R}_+^{H \times W}$ where $H$ and $W$ are the height and with of the depth map.

Unsupervised monocular depth completion relies on photometric (image) and sparse depth reconstruction errors as the primary supervision signal. To this end, we assume an input of $(I_t, z_t)$ for an RGB image and associated sparse depth map captured at time $t$ and during training, an additional set of temporally adjacent images $I_\tau$ for $\tau \in \Upsilon \doteq \{t-1, t+1\}$. The reconstruction $\hat{I}_{t\tau}$ of $I_t$ from another image $I_\tau$ is given by the reprojection based on estimated depth at $\hat{d}_t := f_\theta(\cdot)$

$$\hat{I}_{t\tau}(x) = I_\tau(\pi g_{\tau t} K^{-1} \bar{x} \hat{d}_t(x)) \tag{1}$$

where $f_\theta$ denotes a depth completion network if it takes both image and sparse depth map $(I_t, z_t)$ as input, and a monocular depth estimation network if just $I_t$; $\bar{x} = [x^\top, 1]^\top$ are the homogeneous coordinates of $x \in \Omega$, $g_{\tau t} \in SE(3)$ is the relative pose (rotation and translation) of the camera from time $t$ to $\tau$, $K$ the intrinsic calibration matrix, and $\pi$ is a canonical perspective projection.

Using Eqn. 1, a depth completion network $f_\theta$ is trained by

$$\arg\min_\theta \sum_{\tau \in \Upsilon} \sum_{x \in \Omega} \alpha \rho\big(\hat{I}_{t\tau}(x), I_t(x)\big) + \sum_{x \in \Omega_z} \beta \psi\big(\hat{d}_t(x), z_t(x)\big) + \lambda R(\hat{d}_t) \tag{2}$$

where $\rho$ denotes the photometric reconstruction error, typically $L_1$ difference in pixel values and/or structural similarity (SSIM), $\psi$ the sparse depth reconstruction error, typically $L_1$ or $L_2$, $R$ the regularizer that biases the depth map to be piece-wise smooth with depth discontinuities aligned with edges in the image, and $\alpha$, $\beta$ and $\lambda$ their respective weighting. Note that monocular depth estimation methods share a similar loss, except without the sparse depth reconstruction term and often computes the photometric term based on the minimum error (Godard et al., 2019).

Let $\mathcal{A}_{pt}$ be the set of possible photometric transformations, and $\mathcal{A}_{ge}$ be the set of all geometric transformations, such that $\forall\, T_{ge} \in \mathcal{A}_{ge}$, there exists an inverse transformation $T_{ge}^{-1}$ where $T_{ge} \circ T_{ge}^{-1} \approx Id$ the identity function. Note that since almost all geometric transformations defined on the space of quantized images with finite discrete height and width are not one-to-one, and hence clearly for which no true inverse exists, we have to relax our definition of the inverse transformation. Formally, given geometric transform $T_{ge} \in \mathcal{A}_{ge}$, we define

$$T_{ge}^{-1} = \arg\inf_{T \in \mathcal{A}_{ge}} \{\delta_{ge}(T_{ge} \circ T, Id)\} \tag{3}$$

where $\delta_{ge}$ is a metric on $\mathcal{A}_{ge}$ and $Id$ is the identity mapping. We provide several examples in Sec. 4. At each time step, we can sample a sequence of transformations $\{T_{pt}^1 \dots T_{pt}^k\}$ and $\{T_{ge}^1 \dots T_{ge}^m\}$

---

**Algorithm 1** AUGUNDO FOR UNSUPERVISED DEPTH COMPLETION

**Require:** Depth network $f_\theta$, Images $I_t, I_\tau$, Sparse depth $z_t$, Relative pose $g_{rt}$, Intrinsics $K$
1: Sample $\{T_{pt}^1 \dots T_{pt}^k\}$ from $T_{pt}^i \in \mathcal{A}_{pt}$, and compose $T_{pt} = T_{pt}^1 \circ T_{pt}^2 \circ \cdots \circ T_{pt}^k$
2: Sample $\{T_{ge}^1 \dots T_{ge}^m\}$ from $T_{ge}^i \in \mathcal{A}_{ge}$, and compose $T_{ge} = T_{ge}^1 \circ T_{ge}^2 \circ \cdots \circ T_{ge}^m$
3: Compose the inverse geometric transform $T_{ge}^{-1} = (T_{ge}^m)^{-1} \circ (T_{ge}^{m-1})^{-1} \circ \cdots \circ (T_{ge}^1)^{-1}$
4: Augment $I_t$ to obtain $I_t' = T_{ge} \circ T_{pt}(I_t)$ and $z_t$ to obtain $z_t' = T_{ge} \circ z_t$ (Eqn. 4, 5)
5: Obtain depth prediction $\hat{d}_t' = f_\theta(I_t', z_t')$
6: Apply inverse geometric transformation on output depth map: $\hat{d}_t = T_{ge}^{-1} \circ \hat{d}_t'$ (Eqn. 6, 7)
7: Reconstruct $I_t$ from $I_\tau$ using Eqn. 1, i.e., $\hat{I}_{t\tau} = I_\tau(\pi g_{\tau t} K^{-1} \bar{x} \hat{d}_t)$
8: Minimize reconstruction losses between $\hat{I}_{t\tau}$ and $I_t$, and $\hat{d}_t$ and $z_t$, and the regularizer (Eqn. 8)

---

respectively from $\mathcal{A}_{pt}$ and $\mathcal{A}_{ge}$ to construct transformations $T_{pt} = T_{pt}^1 \circ \cdots \circ T_{pt}^k$ and $T_{ge} = T_{ge}^1 \circ \cdots \circ T_{ge}^m$. We denote the composition of a collection of augmentation transformations by $T = T_{ge} \circ T_{pt}$ where $T_{pt}$ denotes photometric transformation, and $T_{ge}$ denotes geometric transformation. Furthermore, we denote the inverse geometric transformations by $T_{ge}^{-1} = (T_{ge}^m)^{-1} \circ (T_{ge}^{m-1})^{-1} \circ \cdots \circ (T_{ge}^1)^{-1}$, which operates on the space of depth maps to reverse the geometric transformation so that we can warp the output depth map onto the reference frame of the original image. Specifically, each geometric transformation is performed over the coordinates of the image and resampled:

$$[x' \quad 1]^\top = T_{ge} [x \quad 1]^\top \tag{4}$$

$$I'(x') = I(x); \; z'(x') = z(x) \tag{5}$$

where $T_{ge}$ is the geometric transformation, $x \in \Omega$ and $x' \in \Omega$ are coordinates in the image grid, and $I'$ is the image after the transformation; for ease of notation, we hereafter denote $I' = T(I) = T_{ge} \circ T_{pt}(I_t)$ to include the photometric augmentation and inverse warped through the composition. Note that $x$ is in the original image reference frame and $x'$ is in the transformed image reference frame. Naturally, this process can be extended to multiple geometric transformations by composing them i.e. $T_{ge} = T_{ge}^1 \circ T_{ge}^2 \circ \cdots \circ T_{ge}^m$. The reverse process is simply inverting the transformations where $T_{ge}^{-1} = (T_{ge}^m)^{-1} \circ (T_{ge}^{m-1})^{-1} \circ \cdots \circ (T_{ge}^1)^{-1}$:

$$[x'' \quad 1]^\top = T_{ge}^{-1} [x' \quad 1]^\top \tag{6}$$

$$\hat{d}(x'') = \hat{d}'(x') \tag{7}$$

where $\hat{d}'$ is the depth map inferred from transformed image $I'$ and sparse depth map $z'$. Once reverted back to the original reference frame, $\hat{d}$ can be used to reconstructi $I_t$ from $I_\tau$ for $\tau \in \Upsilon$ in Eqn. 1.

By modeling $T_{ge}^{-1}$, Eqn. 4-7 allow us to apply a wide range of augmentations, while still establishing the correspondence between $I_t$ and $I_\tau$. Specifically, for minimizing the loss function (Eqn. 8), one may simply augment the input image by $T$ and feed the augmented image and sparse depth $(I_t', z_t')$ to the depth completion network as input while reconstructing the original image and sparse depth $(I_t, z_t)$ from other views $I_\tau$ and the aligned output depth $\hat{d}$ (Eqn. 7). We note that the inverse transformation is critical for enabling the sparse depth reconstruction term to be computed properly in Eqn. 8; if computed in the transformed reference frame i.e. on $z_t'$, many of the sparse points would be decimated by interpolation during rotation and resizing (downsampling) augmentations – leaving a lack of supervision on sparse depth in the loss function.

As the inverse transformation $T_{ge}^{-1}$ is only a pseudo-inverse, border regions of the image that were out of frame, i.e. translated or cropped away by $T_{ge}$, cannot be recovered. Hence, when we are reversing the geometric transformations on our output depth map, border extensions (edge paddings) are introduced to handle out-of-frame regions. To reduce training noise in these ill-posed regions, we introduce a validity map $\mathbf{1}_t$ to keep track of the padded region and filter out the invalid pixels during loss computation; the data fidelity terms are masked by $\mathbf{1}_t$ and the loss is modified as follows:

$$\arg\min_\theta \sum_{\tau \in \Upsilon} \sum_{x \in \Omega} \alpha \cdot \mathbf{1}_t(x)\rho\big(\hat{I}_{t\tau}(x), I_t(x)\big) + \sum_{x \in \Omega_z} \beta \cdot \mathbf{1}_t(x)\psi\big(\hat{d}_t(x), z_t(x)\big) + \lambda R(\hat{d}_t) \tag{8}$$

for $\mathbf{1}_t \in \{0, 1\}$, where border extended regions are assigned 0 and otherwise 1.

## 4 AUGMENTATIONS

**Photometric.** We include brightness, contrast, saturation, and hue, where all follow standard augmentation pipeline in existing works (Ma et al., 2019; Wong et al., 2020; Wong & Soatto, 2021). The inverse of photometric augmentation of can be viewed as original image before augmentation.

**Occlusion.** We consider image patch removal and sparse point removal. For image patch removal, we randomly select a percentage of pixels $x \in \Omega$ in the image and remove an arbitrary-sized patch around it by setting it to 0. For sparse point removal, we randomly sample a percentage of points $x \in \Omega_z$ in the sparse depth map and set them to 0. Note: we are the first to use and explore photometric occlusion augmentation in monocular depth completion. The inverse transformation of this is simply the original image and sparse depth map before augmentation, both used in loss computation.

**Flip.** We consider horizontal and vertical flips. When applied, the same flip operation are used for both input image and sparse depth maps. We record the flip type during data augmentation. During loss computation, we flip the output depth to align with the original image and sparse depth map.

**Resize.** We define a new image plane of the same dimensions as the input and generate a random scaling factor to be applied along both height and width directions. The image is warped to the new image plane, where any point warped out of the plane is excluded; borders of the warped image are extended to the bounds of the image plane by edge replication. We record the scaling factors during augmentations. During loss computation, we warp the output depth map onto a new image plane of the same dimensions by the inverse scaling matrix of the recorded scaling factors; borders of the warped depth map are extended to the bounds of the image plane by edge replication. Extended regions are assigned 0s (otherwise 1s) in validity mask $\mathbf{1}_t$ to be used in Eqn. 8. Note: we consider resize factors greater and less than 1 as separate augmentations to study their individual effect.

**Rotation.** A naive implementation of random rotation leads to loss of large areas of the image, i.e., cropped away to retain the image lattice. In order to preserve the entire image, we first randomly generate an angle of rotation, define a new (larger) image plane, and then resample the image based on the rotated coordinates such that the rotated image fits tightly within the new image. As image sizes within a batch can vary depending on the rotation angle, we center-pad each image in the batch to the maximum width and height of the augmented batch so that one can construct a batch with the same image dimension. To reverse the rotation on the output depth, we warp the output depth map back by the inverse rotation matrix using the recorded the angle during augmentation. We perform a center crop on the depth map to align with the original image and sparse depth map.

**Translation.** We define a new image plane of the same dimensions as the input and generate two random translation factors along the height and width directions. The coordinates of the input are translated and its pixels are inverse warped onto the new image plane. Any pixel warped out of the image plane is excluded. Borders of the warped image are extended to the bounds of the image plane by edge replication. We record the translation factors during data augmentations; during loss computation, we warp the output depth map onto a new image plane of the same dimensions by the inverse translation matrix and borders of the warped depth map are extended to the bounds of the image plane by edge replication. Extended regions are assigned 0s in validity mask $\mathbf{1}_t$ (Eqn. 8).

## 5 EXPERIMENTS

We demonstrate our method on three recent unsupervised depth completion methods (VOICED (Wong et al., 2020), FusionNet (Wong et al., 2021a), and KBNet (Wong & Soatto, 2021)) using two standard benchmark datasets (KITTI (Geiger et al., 2012; Uhrig et al., 2017), VOID (Wong et al., 2020)). Additionally, we show that our method is also applicable towards monocular depth estimation, which does not utilize the sparse depth modality, but are trained in a similar fashion by minimizing a form of photometric reconstruction loss; we demonstrate our approach on Monodepth (Godard et al., 2019), HRDepth (Lyu et al., 2021), and LiteMono (Zhang et al., 2023a). Descriptions of KITTI and VOID datasets can be found in section A.3 of the Appendix.

To test the zero-shot generalization improvements obtained by training with AugUndo, we test on two additional datasets for both depth completion and depth estimation: NYUv2 (Silberman et al., 2012) and ScanNet (Dai et al., 2017) – where we test the generalization of models trained on VOID.

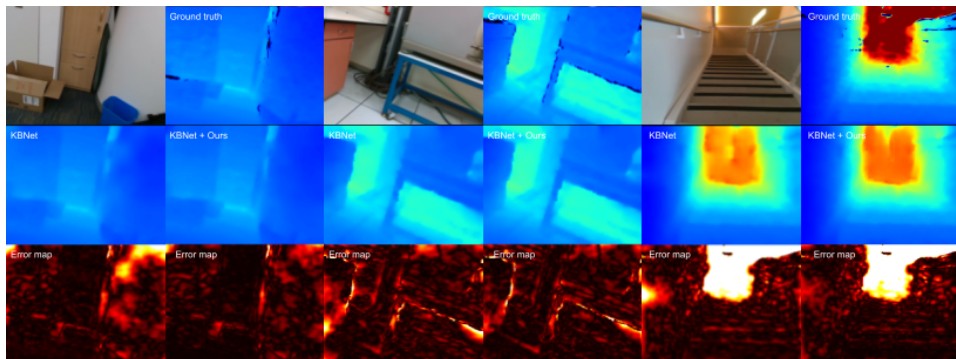

Figure 2: *Qualitative comparisons on VOID*. We compare KBNet trained using standard augmentations and KBNet trained with AugUndo. AugUndo consistently produces lower errors with reduced border artifacts and homogeneous regions i.e., pillar (left), cabinet (middle), wall of staircase (right).

| Method | MAE ↓ | RMSE ↓ | iMAE ↓ | iRMSE ↓ |
|---|---|---|---|---|
| VOICED | 74.78±2.69 | 139.75±4.57 | 39.20±1.46 | 71.98±2.54 |
| + AugUndo | **53.31**±0.54 | **111.87**±1.56 | **27.13**± 0.39 | **54.62**± 0.58 |
| FusionNet | 52.11±0.44 | 113.30±1.18 | 28.53±0.52 | 58.79±2.01 |
| + AugUndo | **41.43** ± 0.46 | **99.74** ± 0.91 | **22.58** ± 0.39 | **54.11** ± 1.78 |
| KBNet | 38.11 ± 0.77 | 95.22 ± 1.72 | 19.51 ± 0.14 | 46.70 ± 0.48 |
| + AugUndo | **33.45** ± 0.27 | **85.85** ± 0.17 | **16.79** ± 0.18 | **41.52** ± 0.39 |

Table 1: *Quantitative results of depth completion on VOID*. Reported scores are mean and standard deviation over four independent trials. "+ AugUndo" improves performance by an average of 17.66% across all methods and evaluation metrics.

In addition, for depth estimation, we further test on Make3d (Saxena et al., 2008) dataset. Details on Make3D, NYUv2, and ScanNet can be found in section A.3 of the Appendix.

**Experiment setup.** Code of each work is obtained from the respective Github repository; we modified their data handling and loss function to incorporate AugUndo. For each setting, we perform 4 independent trials for each model and report the means and standard deviations over all trials. To ensure fair comparison, we train all models from scratch. All models were trained following the author-specified settings. For augmentations, we experimented with a range of combinations both in the type and values chosen. Below, we report the best combination through extensive experiments on each dataset. Evaluation metrics for each model are detailed in Table 5 in the Appendix.

## 5.1 RESULTS ON VOID

**Augmentations.** Through a search over augmentation types and values, we found a consistent set of augmentations that tends to yield improvements across all methods with small changes to degree of augmentation catered to each method. For *depth completion*, we applied photometric transformations of random brightness from 0.5 to 1.5, contrast from 0.5 to 1.5, saturation from 0.5 to 1.5, and hue from -0.1 to 0.1. We applied image patch removal by selecting between 0.1% to 0.5% of the pixels and removing $5 \times 5$ patches centered on them – approximately removing between 2.5% to 12.5% of the image. We applied random sparse depth point removal at a rate between 60% and 70% of all sparse points. We further applied geometric transformations of random horizontal and vertical flips, up to 10% of translation and between -25 to 25 degrees of rotation. For KBNet and FusionNet, we applied random resize factor between 0.6 to 1, while for VOICED, we applied a resize factor between 0.7 to 1. We found that resize factor larger than 1 consistently yielded worse results. For *depth estimation*, we applied photometric transformation of random brightness from 0.5 to 1.5, contrast from 0.5 to 1.5, saturation from 0.5 to 1.5, hue from -0.1 to 0.1. We applied geometric transformation of random rotation between -10 to 10 degrees and random horizontal flipping. We further applied random resize factor between 0.8 to 1. Note that augmentations are applied with a 50% probability. Rows in Table 1, 2 marked with "+ AugUndo" denotes methods trained with our augmentation scheme.

**Results.** Table 1 shows our main results on the VOID depth completion dataset. By training the models with our augmentation strategy, we observe an average overall improvement of 17.66% on

| Method | MAE ↓ | RMSE ↓ | Abs Rel ↓ | Sq Rel ↓ | $\delta < 1.25$ ↑ | $\delta < 1.25^2$ ↑ | $\delta < 1.25^3$ ↑ |
|---|---|---|---|---|---|---|---|
| Monodepth2 | 283.861 ± 3.732 | 395.947± 5.728 | 0.183± 0.002 | 0.100 ± 0.003 | 0.717 ±0.005 | 0.922 ± 0.004 | 0.975± 0.002 |
| + AugUndo | **277.696**±4.861 | **388.088**±5.768 | **0.178**±0.003 | **0.095**±0.004 | **0.724** ±0.007 | **0.925**±0.004 | **0.978**±0.002 |
| HR-Depth | 286.282± 7.059 | 399.112± 9.184 | 0.185± 0.004 | 0.100± 0.004 | 0.714 ± 0.012 | 0.919 ± 0.006 | 0.975± 0.002 |
| + AugUndo | **283.086**±6.787 | **394.261**±9.133 | **0.181**±0.005 | **0.097**±0.005 | **0.718**±0.013 | **0.922**±0.004 | **0.977**±0.002 |
| Lite-Mono | 319.910±15.00 | 446.005±22.97 | 0.209±0.013 | 0.129±0.019 | 0.669±0.016 | 0.892±0.011 | 0.963±0.006 |
| + AugUndo | **308.010**±0.859 | **426.626**±0.484 | **0.200**±0.003 | **0.114**±0.001 | **0.674**±0.005 | **0.901**±0.002 | **0.969**±0.002 |

Table 2: *Quatitative results of monocular depth estimation on VOID*. AugUndo can also provide monocular depth estimation models with consistent boost in performance.

the VOID dataset across all methods and metrics. Specifically, we improve VOICED by 25.89%, FusionNet by 15.32%, and KBNet by 11.77%. This experiment validates our conjecture that by applying a wider range of data augmentation, we are able to improve the baseline model's performance. This also illustrates the lack in use of augmentations in existing works: incorporating standard augmentations (albeit requires modification to the standard augmentation and loss computation pipelines) can yield a large performance boost. Figure 2 shows a head-to-head comparison between KBNet trained using standard procedure in Wong & Soatto (2021) and KBNet trained using our augmentation scheme. We observe qualitative improvements from using our augmentation scheme where we outperform KBNet, i.e., pillar (left), cabinet (middle), wall (right). Including the geometric augmentations consistently yields fewer border artifacts as we translate part of the image out of frame but computes the loss on the original frame of reference through our "pseudo"-inverse transformations of the depth map. This allows models to learn the occluded regions due to motion (i.e. forward), where the borders of the image is missing in the standard training scheme, resulting in failures recover structures near the image border. Here, we show that translation can model this effect in the input space, but with a loss computed without the induced occlusion.

We note that one of the limitations in existing augmentations lies in that there are little to no augmentations applied to sparse depth modality. Here, we include geometric and occlusion augmentations made possible by AugUndo and greatly increase data variations to avoid overfitting to sparse depth. An ablation study demonstrating their importance can be found in the Appendix.

Additionally, AugUndo is applicable for training monocular depth networks as well. Table 2 shows a comparison between the standard augmentation procedure of Monodepth2, HR-Depth, and Lite-Mono and AugUndo. AugUndo consistently improves all models across all error and accuracy metrics. Specifically, we observe a boost in the most difficult accuracy metric ($\delta < 1.25$), where Monodepth2 improves from 0.717 to 0.724, HR-Depth from 0.714 to 0.718 and Lite-Mono from 0.669 to 0.674.

## 5.2 RESULTS ON KITTI

**Augmentations.** For *depth completion*, we applied photometric transformations of random brightness, random contrast, random saturation from 0.5 to 1.5 and random hue from -0.1 to 0.1. We applied image patch removal by selecting between 0.1% to 0.5% of the pixels and removing $5 \times 5$ patches centered on them. We applied random sparse depth point removal at a rate between 60% and 70% of all sparse points. We further applied random horizontal flips, up to 10% of translation and between -20 to 20 degrees of rotation. We found that vertical flips, and both resizing operations are detrimental performance. For *depth estimation*, we applied the same photometric transformation as depth completion. We also applied geometric transformation of random horizontal flips, up to 30% of random translation and between -30 to 30 degrees of random rotation. We found consistent performance degradation with certain augmentations, the details of which can be found in Appendix.

**Results.** While AugUndo consistently improves all methods (Table 3), we note that the improvement is much less in this case: ≈3.18% overall with the largest gain in FusionNet of 5.04%. This is largely due to the small scene variations in the outdoor driving scenarios, i.e. ground plane, vehicles and buildings ahead and on the sides, horizontal lidar scans, and largely planar motion. The dataset bias is strong enough to render vertical flip and resize augmentations to be detrimental to the performance.

Table 4 shows AugUndo for *depth estimation*. We observe similar trends in performance gain: Applying our set of augmentations improve most metrics for all methods. For Monodepth2, all metrics improves except for Sq Rel, and maximum improvement is observed with MAE and Abs Rel at around 3%. For HR-Depth, our methods improves all metrics, while having less improvement

| Method | MAE ↓ | RMSE ↓ | iMAE ↓ | iRMSE ↓ |
|---|---|---|---|---|
| VOICED | 318.59±7.74 | 1,213.60±17.49 | 1.30±0.05 | 3.72±0.04 |
| + AugUndo | **315.06**±1.63 | **1,188.68**±6.74 | **1.28**±0.00 | **3.62**±0.02 |
| FusionNet | 285.55±2.16 | 1,174.47±10.67 | 1.20±0.03 | 3.45±0.08 |
| + AugUndo | **270.06**±1.40 | **1,161.05**±8.29 | **1.11**±0.02 | **3.24**±0.04 |
| KBNet | 263.90±3.63 | 1,130.66±6.22 | 1.05±0.01 | 3.24±0.04 |
| + AugUndo | **256.65**±0.88 | **1,116.53**±4.96 | **1.01**±0.01 | **3.15**±0.06 |

Table 3: *Quatitative results of depth completion on KITTI.* AugUndo consistently improves performance across all methods for all evaluation metrics, notably, by as much as 5% overall for FusionNet.

| Method | MAE ↓ | RMSE ↓ | Abs Rel ↓ | Sq Rel ↓ | $\delta < 1.25$ ↑ | $\delta < 1.25^2$ ↑ | $\delta < 1.25^3$ ↑ |
|---|---|---|---|---|---|---|---|
| Monodepth2 | 2.315 ± 0.005 | 4.794 ± 0.035 | 0.117± 0.001 | **0.845** ± 0.030 | 0.869± 0.004 | 0.959 ± 0.001 | **0.982**± 0.001 |
| + AugUndo | **2.237** ± 0.014 | **4.739** ± 0.032 | **0.113** ± 0.000 | 0.862 ± 0.030 | **0.879** ± 0.002 | **0.960** ± 0.001 | **0.982** ± 0.001 |
| HR-Depth | 2.226 ± 0.004 | 4.626 ± 0.032 | 0.113± 0.001 | 0.797 ± 0.022 | 0.879 ± 0.002 | 0.961± 0.001 | 0.982 ± 0.000 |
| +AugUndo | **2.185** ±0.013 | **4.610**±0.029 | **0.111** ± 0.001 | **0.794** ±0.021 | **0.883**± 0.001 | **0.962**± 0.001 | **0.983**± 0.001 |
| Lite-Mono | 2.338 ± 0.005 | 4.821±0.027 | 0.121 ± 0.001 | 0.875± 0.007 | 0.862± 0.002 | **0.955**± 0.001 | 0.980± 0.000 |
| + AugUndo | **2.314** ± 0.030 | **4.780**± 0.055 | **0.120**± 0.001 | **0.849**± 0.019 | **0.863**± 0.004 | **0.955**± 0.001 | **0.981**± 0.001 |

Table 4: *Quatitative results of monocular depth estimation on KITTI.* AugUndo improves upon most of the metrics across different models consistently.

per metric as compared to Monodepth2. For Lite-Mono, we also improved all metrics, with a 3% improvement in Sq Rel. Qualitative comparisons can be found in Figure 3 of the Appendix

Nonetheless, the performance improvements are obtained nearly for free as these AugUndo only add negligible time to training. The percentage gain, however, is similar to that of innovating a new method, i.e. architectural innovation, such as the improvement of Lite-Mono (Zhang et al., 2023a) over HR-Depth (Lyu et al., 2021), and that of HR-Depth over PackNet (Guizilini et al., 2020).

## 6 DISCUSSION

Conventionally, data augmentation aims to seek visual invariance and create a collection of equivalent classes. One prominent example is contrastive learning, where two samples, augmented from the same source image, are treated as a positive pair, and the network endeavors to correctly identify them as representing the same image. Conversely, when dealing with geometric tasks and employing geometric transformations as augmentations, the underlying objective differs. The same object in distinct poses exhibits entirely distinct geometric properties and should never be regarded as equivalent from a geometric perspective. This lack of equivalence has led to the under-exploration of geometric augmentation. We by-pass this obstacle by un-doing augmentation in the training loss.

On geometric tasks, geometric augmentation holds significant importance because current neural networks typically have no awareness of rules in the physical world such as geometric priors and constraints. Our approach explicitly incorporates these rules derived from the physical world, thereby directly embedding them into the data. We believe this is a key factor in solving geometric tasks. Our experiments validate this perspective, as geometric augmentation consistently enhances the performance of different baseline methods across tasks.

Despite the extensive adoption of data augmentation in computer vision, determining when and how to employ augmentations remains ambiguous and highly dependent on the specific computer vision task and network models. The inclusion of additional augmentation techniques exponentially expands the amount of available training data. Recent interest in seeking generic visual intelligence through massive amounts of training data again brings interest in scaling up data augmentation. Therefore, a systemic protocol of properly selecting augmentations would add substantial value to computer vision research. In this work, we choose geometric tasks to approach this goal, since geometric tasks suffer more from insufficiency in annotated data.

Presently, optimal data augmentation is not only determined by the task but also *de facto* treated as a function of the neural network and determined through trial and error by the model's creator. Here we offer an alternative perspective: the optimal model, on the other hand, also be designed as a function of a set of proven viable data augmentations, with robustness under augmentations being a crucial metric to consider. The exploration of this concept remains an open question for future investigation.

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

# A APPENDIX

## A.1 IMPLEMETATION DETAILS

For monocular depth, we implement our approach in PyTorch according to Godard et al. (2019). Specifically, we implemented our augmentation pipeline into the codebases of Godard et al. (2019), Lyu et al. (2021), and Zhang et al. (2023a). The models are optimized using Adam (Kingma & Ba, 2015) with $\beta_1 = 0.9$ and $\beta_2 = 0.999$. We used a input batch size of 12 and the input resolution for KITTI dataset is $640 \times 192$ while the resolution for VOID is $448 \times 256$. We trained the models for 20 epochs with an initial learning rate of $1 \times 10^{-4}$ and drop the learning rate to $1 \times 10^{-5}$ at the 15th epoch. The smoothness loss weight for KITTI is set to 0.001 as per Godard et al. (2019) and 0.01 for VOID as VOID contains more indoor scenes with homogeneous surfaces.

Similarly, for depth completion, we implemented our augmentation pipeline into the codebases of Wong et al. (2020), Wong et al. (2021a), and Wong & Soatto (2021). The models are optimized using Adam (Kingma & Ba, 2015) with $\beta_1 = 0.9$ and $\beta_2 = 0.999$. For VOID, we used an input batch size of 12, and a random crop size of $416 \times 512$. We trained each model for 40 epochs with base learning rate of $1 \times 10^{-4}$ for 20 epochs and decreased to $5 \times 10^{-5}$ for the last 20 epochs.

For Monodepth2 and HR-Depth, we initialize the Resnet encoder weight with the Imagenet-pretrained weight downloaded from pytorch, as specified in their Github Repo. However, we cannot locate the pretrained weight used for Lite-Mono throughout their Repo, which left us no choice but to train their model from scratch.

## A.2 EVALUATION METRICS

The evaluation metrics that we use are show in Table 5

| Metric | Definition |
|--------|-----------|
| MAE | $\frac{1}{|\Omega|}\sum_{x \in \Omega}|\hat{d}(x) - d_{gt}(x)|$ |
| RMSE | $\left(\frac{1}{|\Omega|}\sum_{x \in \Omega}|\hat{d}(x) - d_{gt}(x)|^2\right)^{1/2}$ |
| iMAE | $\frac{1}{|\Omega|}\sum_{x \in \Omega}|1/\hat{d}(x) - 1/d_{gt}(x)|$ |
| iRMSE | $\left(\frac{1}{|\Omega|}\sum_{x \in \Omega}|1/\hat{d}(x) - 1/d_{gt}(x)|^2\right)^{1/2}$ |
| AbsRel | $\frac{1}{|\Omega|}\sum_{x \in \Omega}\frac{|\hat{d}(x) - d_{gt}(x)|}{d_{gt}(x)}$ |
| SqRel | $\frac{1}{|\Omega|}\sum_{x \in \Omega}\frac{|\hat{d}(x) - d_{gt}(x)|^2}{d_{gt}(x)}$ |
| Accuracy | % of $z(x)$ s.t. $\delta \doteq \max\left(\frac{z(x)}{z_{gt}(x)}, \frac{z_{gt}(x)}{z(x)}\right) < \text{threshold}$ |

Table 5: *Error metrics. $d_{gt}$ denotes the ground-truth depth.*

## A.3 DATASETS

**KITTI** (Geiger et al., 2013) contains 61 driving scenes with research in autonomous driving and computer vision. It contains calibrated RGB images with sychronized point clouds from Velodyne

lidar, inertial, GPS information, etc. For depth completion (Uhrig et al., 2017), there are $\approx$80,000 raw image frames and associated sparse depth maps, each with a density of $\approx$5%. Ground-truth depth is obtained by accumulating 11 neighbouring raw lidar scans. Semi-dense depth is available for the lower 30% of the image space. We test on the official validation set of 1,000 samples because the online test server has submission restrictions to accomodate multiple trials. For depth estimation, we used Eigen split (Eigen & Fergus, 2015), following Zhou et al. (2017) to preprocess and remove static frames. The remaining training set contains 39,810 monocular triplets and the validation set contains 4,424 triplets. The testing set contains 697 monocular images. We follow the evaluation protocol of Uhrig et al. (2017); Eigen et al. (2014), where output depth is evaluated between 0 to 80 meters.

**VOID** (Wong et al., 2020) comprises indoor (laboratories, classrooms) and outdoor (gardens) scenes with synchronized $640 \times 480$ RGB images and sparse depth maps. XIVO (Fei et al., 2019), a VIO system, is used to obtain the sparse depth maps that contain approximately 1500 sparse depth points with a density of about 0.5%. Active stereo is used to acquire the dense ground-truth depth maps. In contrast to the typically planar motion in KITTI, VOID has 56 sequences with challenging 6 DoF motion captured on rolling shutter. 48 sequences (about 45,000 frames) are assigned for training and 8 for testing (800 frames). We follow the evaluation protocol of Wong et al. (2020) where predicted depth is evaluated between 0.2 and 5.0 meters.

**NYUv2** (Silberman et al., 2012) consists of 372K synchronized $640 \times 480$ RGB images and depth maps for 464 indoors scenes (household, offices, commercial), captured with a Microsoft Kinect. The official split consisting in 249 training and 215 test scenes. We use the official test set of 654 images. Because there are no sparse depth maps provided, we sampled $\approx 1500$ points from the depth map via Harris corner detector (Harris et al., 1988) to mimic the sparse depth produced by SLAM/VIO. We test models trained on VOID to evaluate their generalization to NYUv2.

**ScanNet** (Dai et al., 2017) consists of RGB-D data for 1,513 indoor scenes with 2.5 million images and corresponding dense depth map. Because there are no sparse depth maps provided, we sampled $\approx 1500$ points from the depth map via Harris corner detector (Harris et al., 1988) to mimic the sparse depth produced by SLAM/VIO. We followed Dai et al. (2017) and used 100 scenes (scene707-scene806), for zero-shot generalization for models trained on VOID.

**Make3D** (Saxena et al., 2009) contains 134 test images with $2272 \times 1707$ resolution. Ground-truth depth maps are given at $305 \times 55$ resolution and must be rescaled and interpolated. We use the central cropping proposed by Godard et al. (2017) to get a $852 \times 1707$ center crop of the image. We use standard Make3d evaluation protocol and metrics. We use Make3D to test the generalization of monocular depth estimation models trained on KITTI.

### A.4 QUANTITATIVE RESULTS FOR GENERALIZATION EXPERIMENTS

**Generalization Results on Monocular Depth Completion.** We test our depth completion models trained on VOID directly on NYUv2 dataset and ScanNet dataset. The results are shown in Table 6. Applying our method with VOICED greatly improve the zero-shot generalization ability on both NYUv2 and ScanNet. This is likely due to the scaffolding technique employed by VOICED overfits to the sparse depth configuration of VOID and therefore, does not generalize well when presented with sparse depth map with a different configuration. Augmentations like patch occlusion and geometric transformation introduces variety into the sparse depth, which alleviates overfitting to specific configurations of the sparse depth map. For FusionNet and KBNet, we both see decent improvements in all metrics, which further demonstrates by applying a more diverse set of transformations, we are able to improve generalization, even to new datasets.

**Generalization Results on Monocular Depth Estimation.** For depth estimation, we use the model trained on KITTI for zero-shot evaluation on Make3D, and the model trained on VOID for zero-shot evaluation on NYUv2 and ScanNet. The results are show in 7. Our method generalizes well for Make3D dataset, gaining an average of around 8% improvement over all metrics and all models. For indoor dataset, we also witness improvements to a smaller extent. For Monodepth2, models trained on VOID with our methods has around 3% average improvements when tested on NYUv2 and ScanNet. For HR-Depth and Lite-Mono, slight improvement was observed. We hypothesize that more complex models like HR-Depth and Lite-Mono might require more training and hyper-parameter tuning when there is more data diversity due to the various transformations we perform, and therefore show less improvement either when tested on VOID or tested on other indoor datasets.

| Method | Dataset | MAE ↓ | RMSE ↓ | iMAE ↓ | iRMSE ↓ |
|---|---|---|---|---|---|
| VOICED | | 2240.15±143.90 | 2427.91±143.49 | 211.41±9.60 | 238.99±10.89 |
| + AugUndo | | **998.73**±93.01 | **1185.17**±105.64 | **110.84**±7.77 | **133.29**±8.83 |
| FusionNet | NYUv2 | 132.24±2.12 | 236.16±4.59 | 28.68±0.42 | 61.87±1.20 |
| + AugUndo | | **126.25**±4.62 | **230.88**±9.68 | **26.05**±0.55 | **54.41**±1.57 |
| KBNet | | 138.31±5.74 | 257.99±10.36 | 25.48±0.63 | 51.77±0.99 |
| + AugUndo | | **120.74**±4.96 | **232.75**±9.79 | **22.30**±0.51 | **47.70**±1.10 |
| VOICED | | 1562.99±136.79 | 1764.33±146.39 | 270.02±17.25 | 311.02±17.36 |
| + AugUndo | | **640.73**±67.09 | **793.40**±79.86 | **133.57**±9.90 | **172.21**±10.61 |
| FusionNet | ScanNet | 109.47±3.01 | 206.33±6.11 | 55.45±1.56 | 122.52±2.04 |
| + AugUndo | | **101.17**±3.05 | **197.00**±7.31 | **46.17**±1.85 | **100.81**±7.69 |
| KBNet | | 103.05±4.99 | 217.12±13.35 | 36.23±1.12 | 76.55±2.90 |
| + AugUndo | | **84.80**±7.90 | **178.07**±16.06 | **30.07**±1.51 | **64.89**±2.75 |

Table 6: *Quantative results of zero-shot transfer from VOID to NYUv2 and ScanNet.* Reported scores are mean and standard deviation over four independent trials. AugUndo improves generalization of models trained on VOID to novel datasets such as NYUv2 and ScanNet; overall improvement is

| Method | Dataset | MAE ↓ | RMSE ↓ | Abs Rel ↓ | Sq Rel ↓ | $\delta < 1.25$ ↑ | $\delta < 1.25^2$ ↑ | $\delta < 1.25^3$ ↑ |
|---|---|---|---|---|---|---|---|---|
| Monodepth2 | | - | 7.417 | 0.322 | 3.589 | - | - | - |
| + AugUndo | | 4.109 | **6.803** | **0.272** | **2.769** | 0.606 | 0.848 | 0.936 |
| HR-Depth | Make3D | 4.136 | 6.505 | 0.281 | 2.484 | 0.562 | 0.839 | 0.938 |
| + AugUndo | | **4.023** | **6.428** | **0.272** | **2.393** | **0.584** | **0.848** | **0.94** |
| Lite-Mono | | 5.116 | 8.061 | 0.358 | 4.676 | 0.511 | 0.793 | 0.91 |
| + AugUndo | | **4.728** | **7.518** | **0.321** | **3.887** | **0.529** | **0.817** | **0.926** |
| Monodepth2 | | 0.432 ± 0.003 | 0.556 ± 0.005 | 0.205 ± 0.001 | 0.159 ± 0.001 | 0.683 ± 0.005 | 0.907 ± 0.001 | 0.975 ± 0.001 |
| + AugUndo | | **0.415** ±0.005 | **0.537** ±0.006 | **0.196** ±0.002 | **0.148** ±0.003 | **0.700** ±0.008 | **0.915** ±0.002 | **0.977** ±0.001 |
| HR-Depth | NYUv2 | 0.424 ±0.003 | 0.549 ±0.003 | 0.201 ±0.002 | 0.154 ±0.002 | 0.692±0.003 | 0.910 ±0.002 | **0.976** ±0.001 |
| + AugUndo | | **0.421** ±0.009 | **0.542** ±0.011 | **0.199** ±0.005 | **0.152** ±0.006 | **0.696** ±0.009 | **0.913** ±0.004 | **0.976** ±0.001 |
| Lite-Mono | | 0.480±0.012 | 0.616±0.018 | 0.231±0.008 | 0.199±0.015 | 0.637±0.009 | 0.879±0.009 | 0.964±0.004 |
| + AugUndo | | **0.468**±0.003 | **0.595**±0.003 | **0.225**±0.004 | **0.187**±0.005 | **0.646**±0.002 | **0.889**±0.003 | **0.968**±0.001 |
| Monodepth2 | | 0.284 ±0.004 | 0.368 ±0.005 | 0.177 ±0.002 | 0.097 ±0.002 | 0.741 ±0.006 | 0.931 ±0.003 | 0.980 ±0.001 |
| + AugUndo | | **0.270** ±0.003 | **0.351** ±0.004 | **0.169** ±0.001 | **0.088** ±0.002 | **0.759** ±0.004 | **0.937** ±0.001 | **0.982** ±0.001 |
| HR-Depth | ScanNet | 0.282 ±0.004 | 0.366 ±0.005 | 0.175 ±0.002 | 0.095 ±0.002 | 0.743 ±0.005 | 0.929 ±0.003 | 0.979 ±0.002 |
| + AugUndo | | **0.274** ±0.007 | **0.357** ±0.009 | **0.172** ±0.005 | **0.092** ±0.005 | **0.754** ±0.009 | **0.935** ±0.003 | **0.981** ±0.001 |
| Lite-Mono | | **0.296** ±0.002 | 0.388±0.007 | **0.185**±0.003 | 0.109±0.006 | **0.731**±0.002 | 0.921±0.002 | 0.976±0.001 |
| + AugUndo | | **0.296**±0.001 | **0.382**±0.001 | **0.185**±0.001 | **0.105**±0.002 | 0.728±0.002 | **0.924**±0.001 | **0.977**±0.001 |

Table 7: *Generalization results.* For Make3D, all models are trained on KITTI. Note: for Monodepth2, we use the numbers reported by Godard et al. (2019) and the best trial on KITTI. For NYUv2 and ScanNet, all models are trained on VOID.

## A.5 QUALITATIVE RESULTS FOR MONOCULAR DEPTH ESTIMATION

Applying our methods yields qualitative improvements in the depth prediction. In figure 3, we can see that our method produces a smoother prediction (building on thr right sample) and captures ambiguous regions (vegetation on the left sample) than the baseline method, despite using the same hyper-parameter during training.

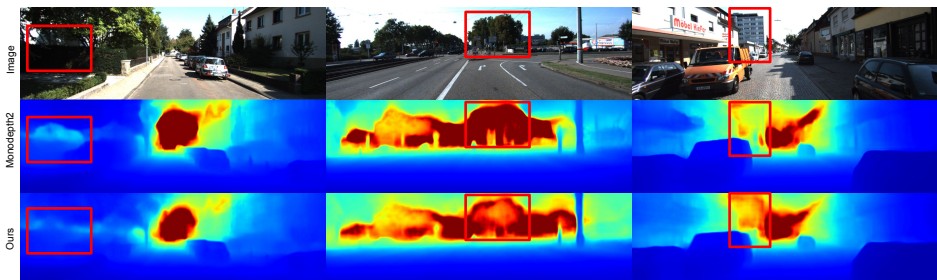

Figure 3: Qualitative result of MonoDepth2 on KITTI. Red bounding boxes highlight areas where training with our augmentation scheme improves Monodepth2, i.e., wall and vegetation on left, trees in middle and building on right.

| Augmentation settings | | | | | | | | Evaluation metrics | | | |
|---|---|---|---|---|---|---|---|---|---|---|---|
| TRN | ROT | HUE | COJ | RMP | FLP | RZD | RMI | MAE | RMSE | iMAE | iRMSE |
| ✓ | ✓ | ✓ | ✓ | ✓ | ✓ | ✓ | ✓ | **33.45**±0.27 | **85.85**±0.17 | **16.79**±0.18 | **41.52**±0.39 |
|  | ✓ | ✓ | ✓ | ✓ | ✓ | ✓ | ✓ | 33.92±0.19 | 87.19±0.55 | 17.09±0.03 | 42.23±0.23 |
| ✓ |  | ✓ | ✓ | ✓ | ✓ | ✓ | ✓ | 33.68±0.19 | 85.86±0.51 | 16.92±0.03 | 41.47±0.18 |
| ✓ | ✓ |  | ✓ | ✓ | ✓ | ✓ | ✓ | 33.60±0.16 | 86.47±0.69 | 16.95±0.27 | 41.65±0.44 |
| ✓ | ✓ | ✓ |  | ✓ | ✓ | ✓ | ✓ | 34.14±0.37 | 87.26±0.74 | 17.09±0.17 | 42.57±1.33 |
| ✓ | ✓ | ✓ | ✓ |  | ✓ | ✓ | ✓ | 37.73±0.27 | 92.62±0.20 | 19.44±0.18 | 45.47±0.37 |
| ✓ | ✓ | ✓ | ✓ | ✓ |  | ✓ | ✓ | 44.38±0.89 | 106.87±0.89 | 22.91±0.68 | 52.55±0.57 |
| ✓ | ✓ | ✓ | ✓ | ✓ | ✓ |  | ✓ | 35.77±0.43 | 89.88±0.84 | 17.81±0.23 | 42.75±0.38 |
| ✓ | ✓ | ✓ | ✓ | ✓ | ✓ | ✓ |  | 33.64±0.22 | 86.26±0.23 | 16.80±0.13 | 41.60±0.64 |

Table 8: *Ablation study for KBNet on VOID dataset.* TRN stands translation with maximum 10% of the image height and width. RZD stands for resize downsampling with a range of 0.60 to 1. ROT stands for rotation with maximum 25 degree. HUE stands for random hue adjustment of 0.1 to -0.1. COJ stands for color jitter using random brightness, contrast, and saturation with the range of 0.5 to 1.5 and hue of -0.1 to 0.1. RMP stands for random remove points and FLP for flip augmentations. RMI stands for random removal of image patches. FLP has the largest effect; augmentations with the highest influence are remove points, flip, and resize. Best performance is achieved when using all of the proposed augmentations.

## B  ABLATION STUDY OF AUGMENTATIONS

In Table 8, we provide a comprehensive ablation study for various augmentations discussed in the main paper. From our best reported settings on KBNet (Wong & Soatto, 2021), we removed individual augmentations to show their contribution to empirical results for the depth completion task. We perform 4 trials for each augmentation setting and took their average to yield the reported numbers. The ablated settings are as follows: translation (TRN) with maximum 10% of the image height and width, rotation (ROT) with a maximum of 10 degrees, random hue adjustment (HUE) from 0.1 to -0.1, random brightness, contrast, and saturation (COJ) with the range of 0.5 to 1.5 for each, random sparse depth points removal (RMP) between 60% and 70% of total points, and horizontal and vertical flip (FLP) augmentations, resize downsampling (RZD) using factors from 0.6 to 1, random remove image patch (RMI) between 0.1% and 0.5% of pixels using 5 by 5 patches. We also include a setting for removing all photometric augmentations to further quantify its effect.

We found that random flips (FLP, row 6) has the largest impact of all of the augmentations – increasing the error by an average of 30% across all metrics. As there are no "photometric" data augmentations applied to the sparse depth maps, RMP is the only augmentation to increase variability in the sparse points configuration; hence it also has large influence. As the sparse points tracked by visual inertial odometry (VIO) systems can vary even from frame to frame (many of dropped and added to the state), RMP serves as the only augmentation to model this process and thus has a large effect on performance. In our present augmentation scheme, we reconstruct the removed points when we minimize the loss and in fact allow this augmentation to serve as an additional training signal realized in an L1 supervised loss.

Photometric augmentations, individually, have small effect on performance. For example, removing hue (row 3) and all color jitter (row 4) yielded small increases in errors. Similar results were found for other individual photometric augmentations (brightness, contrast, saturation, hue), which we grouped as COJ. When removing COJ, we see a slightly larger effect (row 4). However, the change from adding photometric augmentation is still smaller than geometric augmentations we introduced like translation. This demonstrates the the limtations of existing works in terms of augmentations as color jittering have little effect. It also stresses the importance of geometric augmentations. Admittedly, the most important one (FLP) is currently being employed by all methods, but resize and point removal play a large role. The best results are obtained when using all of the proposed augmentations, demonstrating the importance of scaling up both photometric and geometric augmentations.

| Data augmentation | MAE ↓ | RMSE ↓ | iMAE ↓ | iRMSE ↓ |
|---|---|---|---|---|
| KBNet | 38.94 | 96.17 | 19.81 | 46.65 |
| + Resize (scale = 1.1) | 41.23 | 100.97 | 20.42 | 48.27 |
| + Ours | 33.45 | 85.85 | 16.79 | 41.52 |
| KBNet | 263.90 | 1,130.66 | 1.05 | 3.24 |
| + Resize (scale = 0.9) | 295.78 | 1212.34 | 1.20 | 3.631 |
| + Resize (scale = 1.1) | 307.95 | 1197.92 | 1.301 | 3.67 |
| + VFLIP | 288.86 | 1207.59 | 1.19 | 3.55 |
| + Ours | 256.65 | 1,116.53 | 1.01 | 3.15 |

Table 9: *Negative Results for depth completion on VOID (top) and KITTI (bottom).* Including augmentations such as Resize (scale = 1.1) harms performance on VOID. Vertical flip and resizing in KITTI also reduces performance.

| Data augmentation | MAE ↓ | RMSE ↓ | Abs Rel ↓ | Sq Rel ↓ | $\delta < 1.25$ ↑ | $< 1.25^2$ ↑ | $< 1.25^3$ ↑ |
|---|---|---|---|---|---|---|---|
| Monodepth2 (KITTI) | 2.315 | 4.794 | 0.117 | 0.845 | 0.869 | 0.959 | 0.982 |
| + Resize (scale = 0.8) | 2.274 | 4.737 | 0.115 | 0.843 | 0.876 | 0.96 | 0.981 |
| + Resize (scale = 1.1) | 2.317 | 4.805 | 0.119 | 0.875 | 0.868 | 0.958 | 0.982 |
| + Resize (scale = 1.2) | 2.346 | 4.914 | 0.119 | 0.943 | 0.871 | 0.957 | 0.979 |
| + PR (Max 1%, $5 \times 5$) | 2.336 | 4.882 | 0.119 | 0.931 | 0.871 | 0.957 | 0.98 |
| + PR (Max 1%, $3 \times 3$) | 2.335 | 4.816 | 0.118 | 0.862 | 0.867 | 0.959 | 0.982 |
| Mondepth2 + Ours (KITTI) | 2.232 | 4.741 | 0.113 | 0.884 | 0.879 | 0.96 | 0.982 |
| + Resize (scale = 0.8) | 2.256 | 4.815 | 0.114 | 0.914 | 0.88 | 0.96 | 0.981 |
| Mondepth2 + Ours (VOID) | 274.425 | 382.85 | 0.176 | 0.092 | 0.729 | 0.93 | 0.98 |
| + vertical flip | 290.078 | 402.226 | 0.187 | 0.104 | 0.709 | 0.917 | 0.973 |

Table 10: *Negative Results for monocular depth estimation on KITTI and VOID.* PR stands for random patch removal with max percentage of pixel selected and a patch of specified size removed around them. Ours stands for the best augmentation setting that we use While including augmentation such as Resize (scale = 0.8) alone improves the baseline, when used together with other data augmentations we get little to no improvements.

## B.1 Negative Results

We highlight representative negative results in Table 9 and Table 10.

For depth completion, we found that resizing to a larger image, i.e. factor larger than 1, generally gives worse performance. Note that for KITTI vertical flip also negatively affect performance – likely due to the bias of right-side up camera orientation. For monocular depth estimation, we find resizing with a factor larger than 1 and patch removal, are in general detrimental.

**Random Resizing.** Random resizing changes the scales of the objects in the scene. We conjecture that current models rely on convolutional filters to extract image features, which are inevitably sensitive to scale change. Given such sensitivity in the feature space, self-supervised depth models, trained by cross-frame correspondence, cannot effectively find the visual correspondence.

**Patch removal.** Similar to resizing, randomly removing patches destroys visual correspondence, making the network hard to train. Another observation is that training loss is mostly accumulated at the object boundaries and corresponding depth discontinuities in the later stage of the training. This forces the network to learn to register its estimated depth to object boundaries. When patches are removed, however, image information in the patches is lost, and the objective of the network becomes inpainting missing depth values in the removed patches, which deviates from the original objective of the network. Notably, this augmentation is helpful for depth completion. This is likely because regions with no photometric value may still have sparse depth points, allowing there to be a supervision signal and hence allows for inference under heavy occlusions.

