# OpenReview forum: "AugUndo: Scaling Up Augmentations for Unsupervised Depth Completion"
_ICLR.cc/2024/Conference — ICLR 2024 Conference Withdrawn Submission_

### Official Review · Reviewer_2LRv · 2023-10-15

**Soundness:** 3 good
**Presentation:** 3 good
**Contribution:** 2 fair
**Rating:** 6
**Confidence:** 4

**Summary:**

This paper proposes AugUndo, a method to enable a wider range of (geometric) augmentations for unsupervised depth completion. The author pointed out that typical data augmentations have seen limited adoption for the task of unsupervised depth completion for the following reasons:
1. certain augmentations either reduce co-visible image regions or create image artifacts
2. resampling and interpolation of sparse depth maps will lose information

The work is motivated by applying more geometric augmentations to this task. The main idea is to apply a series of photometric and geometric transformations to predict a depth map, and then reverse these geometric transformations such that the predicted depth map is warped back to the original reference frame. Then, training can proceed with computing the photometric loss as usual.

**Strengths:**

1. This paper is well-motivated. Limiting the data augmentation to a small set of photometric transformations has been a long-standing pain point for unsupervised depth completion. The authors clearly identified this limitation and sought to address this problem.

2. The proposed method is simple, effective, and technically sound. It can augment both the input image and the sparse depth map. The use of this method does not affect the network design and can be applied to many existing works.

3. The paper provides a good set of experiments to support its claim. The authors showed consistent improvements on 3 different networks with 2 different datasets by applying AugUndo. Additional experiments on depth estimation also demonstrate the applicability of the proposed method. The authors also provide a detailed ablation study in supplemental materials.

**Weaknesses:**

1. The 2nd contribution on approximate inverse was not well discussed in the main paper.
2. It seems like most of the geometric augmentations are image space transformations. This also applies to the sparse depth map. Would this still suffer the same problem of losing depth points after resampling and interpolation?
3. As a general augmentation method, the paper could be made stronger by applying it to other geometric learning tasks. The current application to depth completion and depth estimation is a bit too narrow (this is just a suggestion).

**Questions:**

See Weaknesses

---

### Official Review · Reviewer_naag · 2023-10-29

**Soundness:** 3 good
**Presentation:** 3 good
**Contribution:** 3 good
**Rating:** 6
**Confidence:** 4

**Summary:**

This paper proposes a simple yet effective method to successfully incorporate the geometrical augmentations into the UNSUPERVISED DEPTH COMPLETION and achieves consistent improvements under different datasets using various baseline models.

**Strengths:**

The method is simple and reasonable. The experiments results demonstrate the effectiveness of the methods.

**Weaknesses:**

1. While the experiments have proved the effectiveness of the methods, yet, how to choose the set of geometrical transformations is still clear.
2. Besides, the paper may lack theoretical analysis, which makes the paper more like a technique report.
3. The contribution maybe insufficient. The idea of undoing the augmentations is common and trivial in the training of deep learning tasks such as in semantic segmentations. A theoretical analysis is needed to improve the paper contribution.
4. The ablation studies are missing.

**Questions:**

What is the influence of using different kind of geometrical transformations?
Can more geometrical transformation types further improve the results?

---

### Official Review · Reviewer_qT32 · 2023-10-30

**Soundness:** 2 fair
**Presentation:** 2 fair
**Contribution:** 2 fair
**Rating:** 3
**Confidence:** 5

**Summary:**

For data augmentation protocol in the task of unsupervised depth completion or monocular depth estimation, this paper proposes to undo the augmentation applied to the model input via applying inverse augmentation operations to the model output. So the supervision can be performed in the original ground-truth formats. This can avoid errors introduced by e.g. interpolation of sparse depth supervision. After carefully designed and optimized combination of data augmentation, improvements are achieved on indoor datasets (17.66%) and outdoor datasets (3.18%) for depth completion.

**Strengths:**

[Originality]
This paper looks into a problem probably overlooked by previous work, i.e. data augmentation is limited and under-exploited in the task of depth completion. They either followed some default limited protocols or didn't mange to apply it successfully.

[Quality]
The numerical improvement on indoor datasets looks good, even though the gain on outdoor dataset (KITTI) becomes limited;

[Clarity]
Generally well-written (though the texts in method part are a bit hard to read);

[Significance]
This paper shows a combination of carefully designed data augmentation that can be applied to several methods which brings improvements on all of them through extensive experiments.

**Weaknesses:**

Technically speaking, this method is somewhat limited in novelty, e.g. undoing the effects of data augmentation was explored in Digging Into Self-Supervised Monocular Depth Estimation before. They showed improvements in applying supervision in the full-resolution image is better than lower-res downsampled ground-truth. This paper extends it to a broader set of operations. But fundamentally speaking, The ideas of
1) higher-resolution or higher-quality ground-truth can improve the performance;
2) ensuring the data augmentation doesn't break the supervision signal w.r.t the input is important;
are well-known. From that aspect, the technical contribution of this work looks incremental and weakened.

Also,  it looks unclear to me how critical the engineering of different data augmentation hyper-parameters is in the improvement. It's a pity that with such carefully designed augmentation protocols, the resizing augmentation is still limited in fact: resizing factor cannot be larger than 1 in VOID dataset; no resizing operation can work for KITTI dataset. This further weakened the claim.

**Questions:**

1. Have the authors tried applying this to scene flow estimation from sparse point cloud inputs plus images? If that works, it would greatly enhance the significance of this paper.
2. Why does resizing factor greater than 1 harm performance on VOID dataset?

---

### Official Review · Reviewer_WQDt · 2023-10-31

**Soundness:** 2 fair
**Presentation:** 2 fair
**Contribution:** 2 fair
**Rating:** 3
**Confidence:** 4

**Summary:**

The paper introduces a way to exploit various photometric and geometric augmentations for unsupervised depth completion and estimation tasks. Conventional methods for those tasks have been using only simple augmentations (eg., limited photometric transformation, random horizontal flip) due to their task characteristics (for example, random crop and resizing change the effective focal length of inputs and thus are not suitable for the monocular depth estimation task).
This paper proposes to apply diverse augmentations, infer depth, and undo those augmentations on the output to return it back to the original reference frame coordinate (Fig. 1).
The paper applies this idea to two tasks (depth completion and estimation) and shows decent accuracy improvement on public datasets.

**Strengths:**

- Self-contained

  The paper is easy to follow. Pretty much details are self-contained (Fig. 1 and Algorithm 1 for the main idea). The experiment section provides sufficient details on its hyperparameter choices and makes the method transparent and reproducible.

- Decent accuracy improvement

  Table 1 to Table 4 shows that, when applying the proposed idea to baseline models, it brings decent accuracy improvement on both depth completion and estimation tasks on public datasets.

**Weaknesses:**

- Double-edge sword?

  Applying various geometric augmentation lets the network see diverse samples during training. On the other hand, pixels that go out-of-frames after the augmentation will not be used for training even after the undo operation, which reduces the number of supervision signals. How many pixels (in %) go out of the image frames after the augmentation? (One could simply calculate the mean of the validity map in Eq. (8)). Does heavy augmentation hurt the accuracy? How does the accuracy change over the level of augmentation (such as from weak augmentation to strong augmentation)?

- Weak baselines

  The paper adopts their method on three baselines for depth completion (VOICED, FusionNet, KBNet) and depth estimation (Monodepth2, HR-Depth, Lite-Mono). However, those baselines are outdated, and it questions whether the method can bring benefits to any method. For example, one baseline (VOICED) is ranked 144th out of 158 methods on [KITTI depth completion benchmark](https://www.cvlibs.net/datasets/kitti/eval_depth.php?benchmark=depth_completion). It would have been great if the paper demonstrated the effectiveness of the method on recent/top methods on benchmarks.

- Limited evaluation

  The paper evaluates the method on only two datasets (VOID and KITTI). For the monocular depth task, there are other popular datasets such as NYUv2, ScanNet, Make3D. Although the paper uses those datasets for zero-shot evaluation, it would be still good to train the method on one of the datasets (e.g., NYUv2) and compare it with other state-of-the-art methods.

- Different hyperparameter choices on each baseline model

  In the subsection "Augmentations." in Sec 5.1, the paper uses different sets of hyperparameters for each baseline model (KBNet/FusionNet and VOICED). I wonder why it uses different parameters, and if this method is sensitive to the hyperparameter choices. If it's sensitive, this would decrease the practicality of the method.

**Questions:**

- Unclear sentence

   In the 2nd paragraph of the introduction "Zooming an object in the image domain implies a closer distance between the object and the camera, ...". I am not so sure if it's true. Even if zooming an object in the image domain, I think the actual distance between the object and the camera still remains the same, but only the camera focal length changes. I wonder if that's the case or not.

- Details of $R(\hat{d_t})$ in Eq (2)

  I wonder if the paper can provide the details on the regularizer term $R(\hat{d_t})$ in Eq (2).

- Wrong bold-faced

  In Table 8, the best number for iRMSE is in the 3rd row.

---

### Official Review · Reviewer_YqAp · 2023-11-01

**Soundness:** 2 fair
**Presentation:** 3 good
**Contribution:** 2 fair
**Rating:** 3
**Confidence:** 4

**Summary:**

The paper describes several approaches to data augmentation when training monocular depth estimation or completion networks. The evaluation is done on the KITTI and VOID datasets, showing consistent improvement from the proposed data augmentation strategies.

**Strengths:**

Data augmentations are very important in self-supervised pipelines. Doing data augmentation is not easy in depth completion or estimation problems when depth channel should be transformed consistently with the transformation applied to the color or intensity channels. The paper is correct to highlight this issue. In other cases, such as DINO features, for example, data augmentation is the driver underneath learning a useful universal visual representation.

**Weaknesses:**

The paper proposes some augmentation strategies, however, there are references that use non-trivial data augmentation for depth completion or estimation, but they are not cited and not compared against. In particular, the work "ADAADepth: Adapting Data Augmentation and Attention for Self-Supervised Monocular Depth Estimation" from Kaushik et al., IEEE RAL 2021,  uses a less trivial approach than the proposed paper. While in the proposed paper a network is applied only to the augmented image, in the work I just mentioned the network is applied both to initial and to augmented images, and the outputs are required to be consistent. The ADAADepth method contains a number of augmentation strategies and shows good results on KITTI dataset. It is a question why we do not see a comparison to this method in the submission, and why the methods to augment data proposed in the ADAADepth work are not discussed in the submission.

There are also some issues with the mathematical formulation of the method, see below in the Questions section.

**Questions:**

1. Why the reference mentioned above is not cited and is not compared against?

2. The paper says that the method is "Aligning the training target with the augmented input", but in other places it says the method “undoes” the augmentations during the loss computation.” For me these approaches are distinct: either the target is transformed, or the output of the network is de-augmented. Looking at Algorithm 1 I see that most likely the method is not aligning training target, but aligning the output after the augmentation with the training target. The text should be made consistent.

3. Mathematical definition of depth completion is incorrect: The depth completion is not a mapping from \Omega x \Omega_z \to R_+, but a mapping from \Omega \to R_+. We do not put every pixel and every sparse point with depth in correspondence with depth prediction, but we just predict depth for every pixel, so it should be a function from pixels to depth values.

4. What are “geometric transformations”? from (4), it is a transformation defined on the homogeneous pixel coordinates, am I right? Should be explicitly said somewhere; in Algorithm 1, T_ge is applied to z_t, to image as well, but in the formulas T_ge is applied to the coordinates

5. (5) incorrect, I’ is also after a photometric transformation as written in the text

---

### Author Response · Authors · 2023-11-17
**A general message to the reviewers and ACs**

We appreciate the reviewers for their valuable feedback, including typos and unclear notations. We will fix them in the next revision.

**On additional comparisons with Kaushik et al.**: Their method is focused on monocular depth estimation; the authors did not open-source their code and adapting it to depth completion is non-trivial. Their method proposed a novel architecture for depth estimation together with an augmentation method; our method is model-agnostic, which we demonstrate on three different existing depth completion architectures as well as another three existing depth estimation architectures.


**On weak baselines**: The rankings suggested by the reviewer contain both supervised and unsupervised methods. Naturally, supervised methods will dominate, but this does not imply that our choice of unsupervised depth completion models are weak baselines – all are competitive.

**On limited evaluation**: We trained and evaluated on the standard benchmarks of unsupervised depth completion; this is following the convention in the literature. We also performed additional evaluation on ScanNet, NYUv2 and Make3d. All of these are done with 4 trials each, which totals to thousands of hours of GPU time when summed over all of the choices of combinations over 11 different augmentations, different degrees of augmentations, and across 6 models. While there indeed exist many datasets available that one can also try, we believe we have presented sufficient evidence to support our claims.

**On different choices of hyper-parameters between FusionNet/KBNet and VOICED**: VOICED has a scaffolding step which requires at least 3 points in the input sparse depth map. When choosing a large resize factor, it incurs loss on sparse points (introduction, paragraph 3) to the point where there are no points remaining. So, the hyper-parameter is a limitation imposed by the method and not by our choice.

**On higher-resolution or higher-quality ground-truth can improve the performance**: This is tangential to our work as the performance gain in our method does not come from a higher resolution input, but from the diversity of the input, where we simulate nuisance variabilities – training on them renders us robust and more generalizable. Furthermore, our method applies to unsupervised depth completion, which requires no ground-truth.

**On ``ensuring the data augmentation doesn't break the supervision signal w.r.t the input is important...''**: While the reviewer claims that ensuring data augmentation doesn't break the supervision signal is well-known, it is not trivial to achieve this for unsupervised depth completion. Our method addresses the “how”, in the sense that we propose a method to allow augmentations that would break supervisory signal (if applied naively) in unsupervised depth completion to be used to improve model performance and hence novel.

**On application of method to other geometric tasks**: While this is outside the scope of our work as we focus on unsupervised depth completion, our methods can be extended to other geometric tasks by applying augmentations to pairs of images (optical flow, stereo, ego-motion) or sequence of images (multi-view). However, adaptation in implementation might be required to handle caveats specific to different tasks. For example, for frontoparallel stereo, rotation would be inapplicable as stereo pairs are required to be rectified images. Nevertheless, our method opens exciting opportunities for improving a wide range of geometric tasks and we leave this to be addressed by future works.

We want to clarify that our paper is about unsupervised depth completion, in which the main contribution is to preserve accurate supervision signal in sparse depth under geometric augmentations. We include monocular depth estimation results only as a supplement. However, the reviews are mostly focused on the latter so we kindly withdraw from the ICLR submission. Nonetheless, we still thank all the reviewers for their valuable input.